# An Efficient Authenticated Key Agreement Scheme Supporting Privacy-Preservation for Internet of Drones Communications

**DOI:** 10.3390/s22239534

**Published:** 2022-12-06

**Authors:** Chun-Ta Li, Chi-Yao Weng, Chin-Ling Chen, Cheng-Chi Lee, Yong-Yuan Deng, Agbotiname Lucky Imoize

**Affiliations:** 1Program of Artificial Intelligence and Information Security, Fu Jen Catholic University, No. 510, Zhongzheng Road, New Taipei City 24206, Taiwan; 2Department of Information Management, Tainan University of Technology, No. 529, Zhongzheng Road, Tainan City 71002, Taiwan; 3Department of Computer Science and Artificial Intelligence, National Pingtung University, No. 4-18, Min-Sheng Road, Pingtung City 90003, Taiwan; 4Department of Computer Science and Information Engineering, Chaoyang University of Technology, No. 168, Jifeng East Road, Taichung City 41349, Taiwan; 5School of Information Engineering, Changchun Sci-Tech University, Changchun 130012, China; 6Research and Development Center for Physical Education, Health, and Information Technology, Department of Library and Information Science, Fu Jen Catholic University, No. 510, Zhongzheng Road, New Taipei City 24206, Taiwan; 7Department of Computer Science and Information Engineering, Asia University, No. 500, Lioufeng Road, Taichung City 41354, Taiwan; 8Department of Electrical and Electronics Engineering, Faculty of Engineering, University of Lagos, Akoka, Lagos 100213, Nigeria; 9Department of Electrical Engineering and Information Technology, Institute of Digital Communication, Ruhr University, 44801 Bochum, Germany

**Keywords:** authentication, Internet of Drones, privacy preservation, session key agreement, security

## Abstract

In recent years, due to the rapid development of Internet of things (IoTs), various physical things (objects) in IoTs are smart enough to make their own decisions without the involvement of humans. The smart devices embedded in a drone can sense, collect, and transmit real-time data back to the controller from a designated environment via wireless communication technologies. The mobility, flexibility, reliability and energy efficiency of drones makes them more widely used in IoT environments such as commercial, military, entertainment applications, traffic surveillance and aerial photography. In a generalized IoD architecture, we have communications among the drones in a flying zone, among the drones and the control server, and also among the drones and authorized user. IoD still has many critical issues that need to be addressed, such as data access being carried out through a public channel and battery operated drones. To address these concerns in IoD communications, in this paper, an efficient authentication and secure communication scheme with privacy preservation is proposed and it only uses secure one-way hash function and bitwise XOR operations when control server, drone and user mutually authenticate each other. After the successful authentication, both IoD-based participants can agree on a common session key to secure the subsequent communication messages. The widely accepted ProVerif and BAN logic analysis have been used to assure that the proposed scheme is provably secure against existing well-known security attacks and ensures privacy. Finally, a comparative analysis is presented to demonstrate the proposed scheme preserves efficiency when compared to existing competitive schemes.

## 1. Introduction

For the past few years, as information and communication technology (ICT) advances and smart devices increase dramatically, the Internet of Things (IoT) has become a much-talked-about topic among many experts and large ICT companies [1]. Due to its capability to extend the traditional human-to-human network communication connection for fulfilling communication and dialogue between humans and objects, or further to achieve communication and dialogue between objects, these objects can be distinguished into physical objects and virtual objects. The physical objects include sensors, drones, surveillance cameras, smartphones, self-driving vehicles, and smart homes, while the virtual objects include electronic wallets, electronic tickets, and electronic agendas. The strength of using various smart objects in the IoT environment is that they can operate autonomously without human intervention and can be easily integrated into various smart network applications. Especially, drones can be also named unmanned aerial vehicles (UAVs), which is an aircraft that can be controlled remotely or by an onboard computer. Drones can navigate autonomously without human intervention and are comprised of several IoT smart devices, such as light pulse range sensors (laser), radio detection and distance measuring sensors, magnetic field change sensors, sonar range sensors, time of flight sensors, thermal sensors, chemical sensors, and direction sensors. Along with the miniaturization trend of different devices (processors, microcontrollers, sensors, wireless transceivers) inside the drones, it can be seen as a hint that IoT technology makes the drone network or IoDs a part of IoTs [2,3,4,5,6,7,8].

When combined with various sensing elements, location services, wireless transmission reading, content services, and other technologies, many different types of drone applications have been derived. The application scopes of several drones were described and organized, as shown below:For civilian purposes [9]:**i.** For photography purposes: Allowing TV/film producers to take aerial photography in a new manner by using drones, thus enhancing the aerial view to a higher extent.**ii.** For natural disaster assessment and control purposes: After Hurricane Katrina hit the United States in 2005, drones were used for disaster control and assessment to observe which roads were blocked by fallen trees, cars, and road barriers, or to search for missing, injured, and trapped people.**iii.** For emergency response purposes: Like ambulances, drones can be used as portable medical kits which can send medical supplies to emergency units on site, particularly when the emergency site is inaccessible for vehicles. Furthermore, affected by the recent COVID-19 pandemic, drones have been deployed on the streets of Spain and China (mainly Wuhan), to raise people’s awareness of the crisis via cameras and broadcasters, or aerial spraying for disinfection. Furthermore, drones can be used as a means of delivering food and medication to infected patients, aiming to transport tested samples at a higher speed, and reduce human contact.**iv.** For environmental monitoring purposes: Drones can be used to perform tasks of measuring environmental pollution, such as those for air quality measurement and analysis; perform agricultural tasks, such as soil analysis, crop/livestock management/disease, and pest control; perform animal protection tasks, such as nature/wildlife protection/anti-poaching/endangered species protection.For police purposes [10]:**i.** For traffic monitoring purposes: Drones can be used to monitor traffic and accident scenes. For example, the Spanish government has adopted drones to monitor traffic bottlenecks since 2015.**ii.** For criminal-tracking purposes: Drones can be used to monitor crime scenes and prison fugitives. For example, the Ohio State Police Station used a drone to track an escaped prisoner and track him down in 2016.**iii.** For forensic search-and-rescue purposes: Drones can be used to tackle crimes, such as the missing person and murder case of Ms. Tara Grinstead in 2015, for whom Georgia police used a fixed-wing drone called Spectra to search.For military purposes [11]:**i.** For aerial surveillance/reconnaissance purposes: Drones can be deployed in the air to collect intelligence and information and further identify and track the locations of terrorist camps, vehicles, weapons, plants, and improvised explosive devices. For example, Russia collected new drone footage that unveiled how Turkey used artillery operations to attack the Syrian army in 2020.**ii.** For airstrike purposes: As early as 2002, the U.S. military used drones for airstrike missions and then developed them for application with British allies in the global anti-terrorism war. In addition, Israel also made use of drones to conduct airstrikes against military installations/key targets/people in Iraq and Syria on the west coast.**iii.** For drone hijacking purposes: Drone hijacking is mainly achieved via GPS intervention/spoofing, which was used to resolve the conflict in Ukraine and stood up to the threat from the Islamic State until the city of Mosul was finally liberated from the Islamic State in 2017.For criminal attack purposes [12]:**i.** Physical attacks: Drones can easily be used to destroy people’s privacy and threaten their private property by crashing into people or their property intentionally or unintentionally to cause them serious damage. Moreover, some drones can fly as high as 500 m in the air, just like bird strikes, which can cause serious damages to aircrafts in flight.**ii.** Logical attacks: They include spoofing a hotspot of a mobile Wi-Fi network, allowing the victim users to connect and monitor their sensitive messages, such as account passwords and credit card data, or implanting malware into smartphones and mobile devices that are connected to the malicious hotspot. Furthermore, a Raspberry Pi device connected to a drone can also be maliciously coded to intercept or hijack other drones nearby.

From the perspective of security and threat analysis, drone-assisted public safety networks require a stricter manner rather than traditional wireless networks such as wireless sensor networks (WSN) [13] and mobile ad hoc networks (MANET) [14] to restrict the unauthorized collection of images and videos by drones. Though drones carry less information and less power, they can cover a wider range than WSNs and MANETs. As a result, the challenge of drone network security is how to provide communication channels with confidentiality, integrity, availability, authentication, and non-repudiation over the resource constraints and latency constraints of drones. Actually, kinds of technologies for drone operations and their specific properties are being explored and misused for potential attacks including performing terrorist attacks and reconnaissance, tracking specific people, and monitoring certain properties, thus arousing security and privacy concerns. Furthermore, if a drone is out of order and crashes into nearby private houses, public facilities, parked cars, or civilians, it could also lead to casualties and damage to property. On the other hand, drones mainly make use of Wi-Fi, short-range Wi-Fi, Bluetooth, or other wireless devices, such as Bluetooth-connected keyboards, while, if there are inadequate security measures for connection to these devices, such as insecure single factor authentication and easy-to-break typical passwords, the attackers can easily intercept messages and destroy private buildings and public areas.

In plenty of authentication and key agreement (AKA) schemes [15,16,17,18,19,20,21,22,23], symmetric and asymmetric cryptosystems have been proposed to implement a comprehensive authentication on the use of IoT and IoD environments. However, with the resource-constrained nature of drones, it cannot consume a high amount of energy for executing complex cryptographic operations on large datasets and AKA scheme shall be sufficiently lightweight both in terms of computational complexity, communication overhead and memory demand. Turkanović et al. first proposed an IoT-based AKA scheme [24] for WSNs and their scheme is highly efficient as it only uses lightweight hash and bitwise XOR computations. Although it achieves the condition of lightweight authentication, Farash et al. [25] pointed out that their devised scheme is prone to man-in-the-middle attack, node impersonation attack, and additionally does not render nodes anonymity and user traceability. In order to provide better security, Wazid et al. designed a novel AKA scheme [26] for UAV distributed networks. However, the protocol was pointed out by Lei et al. [27] as not being provided to perfect forward secrecy. Meanwhile, Rodrigues et al. [28] designed two methods for the drone communication environment. The first one is modified based on the AKA scheme of Farash et al. [25], allowing for a direct connection between a drone and another one; the second one is modified based on the AKA scheme of Jiang et al. [18], which allows a drone to communicate with another one through a ground control station. However, their AKA schemes fail to resist ephemeral secret leakage (ESL) attacks under the Canetti–Krawczyk (CK) threat model. Recently, Zhang et al. proposed a lightweight AKA scheme [29] with anonymity and untraceability for IoD environments and their AKA scheme can be proven secure under random oracle model. All the drones and the users are registered with a central trusted authority, control server (namely CS) prior to their deployment. By verifying the validation of the transmitted messages, all participants in IoD can ensure mutual authentication and establish a common session key securely. In this paper, we will propose an improved version of Zhang et al.’s scheme that not only provides the same level of security with anonymity and untraceability but also protects the scheme from various known attacks.

In order to achieve the aforementioned security requirements of previous authentication schemes in IoD environments, in this paper, we propose a lightweight mutual authentication and privacy preservation scheme to resist several security attacks and provide a series of important features cited above. The main contributions of this paper are given as follows: (1) In our lightweight authentication scheme, the properties of drone anonymity and drone untraceability can be guaranteed at authentication and key agreement phase when involved participants transmitted messages via a public IoD channel. (2) In comparison with existing IoT-assisted authentication schemes for IoD communications, our proposed scheme can not only maintain the efficiency of computational and computation overheads, but also achieve basic security features mentioned in prior studies. (3) Informal security analysis and BAN logic analysis are performed and ProVerif-based formal security simulation is implemented, to demonstrate that our scheme is secure against various security attacks.

The remainder of the paper is organized as follows. Section 2 presents a new security architecture along with the threat model for IoD communication environments. Section 3 introduces our authentication and key agreement scheme with privacy preserving for IoD communications. The informal security analysis with the formal security verification using the widely accepted ProVerif simulation and BAN logic of the proposed scheme are given in Section 4. An in-depth performance comparison of the proposed scheme with existing IoD authentication schemes is given in Section 5. Finally this paper is concluded in Section 6.

## 2. System Architecture in IoD Communications

In this section, we will illustrate the proposed system architecture for the IoD paradigm. Subsequently we define two adversary models to evaluate its security and usability.

### 2.1. System Model

In terms of the design, the main participants in this paper were control server (CS), the trusted registration authority, users who could access IoD data using mobile devices, some mobile-type drone nodes deployed in the application fields to collect and broadcast data from the fly zone. CS is a trusted unit responsible for registering and issuing unique identifiers and generating secret parameters for users and drones. By deploying drone nodes via CS in fly zones for authority control, these drone nodes can be seen as cluster heads for a specific fly zone, providing an efficient and well-designed communication and authentication mechanism for IoD environments to avoid the single point of failure of traditional single centralized certificate centers. An external user can access certain specific drone nodes in the IoD environment via Internet communication and his/her mobile device, given that he/she is authenticated and authorized by the CR to access these drones. In this paper, the IoD communication and authentication mechanism for IoD applications included three modes, namely CS-to-Drone communication, CS-to-User communication, and User-to-Drone communication. The overall communication architecture diagram of IoD is illustrated in Figure 1.

### 2.2. Threat Model

According to the system architecture shown in Figure 1, drones, mobile users and control servers can communicate with each other and all communications of IoD take place over the public channels. In threat model, we will adopt the widely-used Dolev–Yao (DY) threat model and Canetti–Krawczyk (CK) adversary model. According to the definition of DY model, the communication channel between any two entities is open and insecure, and also the end-point entities are not trusted. An adversary can eavesdrop and collect on the messages exchanged on IoD network, and can also delete or tamper the transmitted messages over public channel. According to the definition of CK model, the mobile device of an $U_{i}$ may be lost or stolen. The system parameters stored in that device can be also extracted by using power analysis attack. Furthermore, an adversary may physically capture some drone node $V_{j}$ and extract the stored parameters in $V_{j}$ with the help of complicated power analysis attack. Therefore, the compromised data will be used to undermine the security of IoD communications such as session key exposure, impersonation attack, replay attack, privacy exposure attack and man-in-the-middle attack etc. Note that CS is a trusted party and it will not be compromised by adversaries.

## 3. The Proposed Scheme

In this section, we propose a new lightweight authentication and key agreement scheme with privacy preservation for IoD communications. The proposed scheme consists of the following four phases: system setup, user registration, drone registration, and authentication and key agreement phase. The details of the proposed scheme are described in the following subsections. The notations used in the proposed scheme are summarized as follows.

$U_{i}$: The *i*th mobile user.$V_{j}$: The *j*th drone.CS: The control server.$ID_{i},PW_{i}$: The identity and password of $U_{i}$.$ID_{j}$: The identity of $V_{j}$.k,MSK: 160 bits secret value and master key of CS.*n*: 160 bits public parameter selected by CS.$T_{U_{i},V_{j},CS}$: The current timestamp of $U_{i}$, $V_{j}$ and CS, respectively.$r_{1},r_{2}$: 160 bits random numbers of $U_{i}$ and $V_{j}$, respectively.$L_{V_{j}}$: An active drone list.h(·): A collision free one-way hash function.$\Delta_{T}$: The maximum time threshold of accepting messages.time: The current time received message.$SK_{ij}$: The common session key shared between $U_{i}$ and $V_{j}$.⊕: The bitwise exclusive OR operation.||: The string concatenation operation.

### 3.1. System Setup Phase

In this phase, CS first generates MSK and *k* as its master key and secret value, respectively. Then, CS chooses a secure one-way hash function $h:{\left\{0,1\right\}}^{*}\;\to \;Z_{n}^{*}$, where *n* is a 160-bits public parameter chosen by CS. Finally, CS saves (MSK,k) secretly and publishes (h(·),n).

### 3.2. User Registration Phase

In this phase, every mobile user $U_{i}$ needs to perform the user registration procedure with CS via a secure channel. The graphical representation of the registration procedure of the user is depicted in Figure 2.

**Step 1.** $U_{i}$ chooses his/her identity $ID_{i}$, password $PW_{i}$ and a random number $r_{U_{i}}\in Z_{n}^{*}$ and computes $PID_{i}=h(ID_{i}||PW_{i}\left|\right|r_{U_{i}})$. Then $U_{i}$ sends the registration request {$ID_{i},PID_{i}$} to CS via a secure channel.**Step 2.** After receiving the registration request from $U_{i}$, CS checks the uniqueness of $U_{i}$’s identity. If the uniqueness of $ID_{i}$ is satisfied, CS computes $A_{i}=h(PID_{i}\left||MSK\right)$ and sends it to $U_{i}$ securely.**Step 3.** After receiving $A_{i}$ from CS, $U_{i}$ computes $B_{i}=A_{i}⊕h\left(PW_{i}\right)$ and stores {$B_{i},r_{U_{i}}$} in the tamper-proof memory, which means that the parameters $B_{i}$ and $r_{U_{i}}$ can be used during the computation, but it is unable to extract them from the mobile device of $U_{i}$.

### 3.3. Drone Registration Phase

In this phase, every drone $V_{j}$ needs to complete the drone registration procedure with CS via a secure channel. The graphical representation of the registration procedure of the drone is depicted in Figure 3.

**Step 1.** CS selects an unique identity $ID_{j}$ for $V_{j}$ and computes $\alpha_{j}=h(ID_{j}\left||k\right)$. Then CS saves ($ID_{j},\alpha_{j}$) in list $L_{V_{j}}$ and sends {$ID_{j},\alpha_{j}$} to $V_{j}$ securely.**Step 2.** After receiving the registration parameters from CS, $V_{j}$ stores $ID_{j}$ and $\alpha_{j}$ in its memory securely.

### 3.4. Authentication and Key Agreement Phase

After registration, $U_{i}$ and $V_{j}$ can communicate with each other and establish a common session key $SK_{ij}=SK_{ji}$ for securing future communications. The graphical representation of the proposed authentication and key agreement phase is depicted in Figure 4.

**Step 1.** $U_{i}$ opens the login portal and inputs his/her identity $ID_{i}$ and password $PW_{i}$ into the mobile device. Then the mobile device retrieves ($B_{i},r_{U_{i}}$) and computes $PID_{i}=h(ID_{i}||PW_{i}\left|\right|r_{U_{i}})$ and $A_{i}=B_{i}⊕h\left(PW_{i}\right)$. Then it randomly generates two 160 bits random numbers $r_{U_{i}}^{new},r_{1}\in Z_{n}^{*}$ and computes $PID_{i}^{new}=h(ID_{i}||PW_{i}\left|\right|r_{U_{i}}^{new})$, $M_{1}=PID_{i}^{new}⊕h(A_{i}\left|\right|T_{U_{i}})$, $M_{2}=h\left(PID_{i}^{new}\right)⊕r_{1}$, $M_{3}=h(PID_{i}^{new}\left|\right|r_{1}\left|\right|T_{U_{i}})$, where $T_{U_{i}}$ is the current timestamp of $U_{i}$. Then $U_{i}$ sends authentication request message {$PID_{i},M_{1},M_{2},M_{3},$$T_{U_{i}}$} to CS via a public channel.**Step 2.** After receiving the authentication request from $U_{i}$, CS checks whether $time-T_{U_{i}}\leq \Delta_{T}$ holds or not. If not, CS rejects the authentication request immediately. Otherwise, CS computes $A_{i}=h(PID_{i}\left||MSK\right)$, $PID_{i}^{new}=M_{1}⊕h(A_{i}\left|\right|T_{U_{i}})$, $r_{1}^{\prime }=M_{2}⊕h\left(PID_{i}^{new}\right)$, and $M_{3}^{\prime }=h(PID_{i}^{new}\left|\right|r_{1}^{\prime }\left|\right|T_{U_{i}})$.**Step 3.** CS checks whether $M_{3}^{\prime }=M_{3}$ holds or not. If yes, CS authenticates the legality of $U_{i}$. Otherwise, CS rejects $U_{i}$’s authentication request. Now, CS randomly assigns an active drone $V_{j}$ in IoD for $U_{i}$ and computes $M_{4}=h(\alpha_{j}\left|\right|T_{CS})⊕r_{1}$, $M_{5}=h(\alpha_{j}\left|\right|r_{1}\left|\right|T_{CS})⊕PID_{i}^{new}$, $M_{6}=h(\alpha_{j}\left|\right|r_{1}||PID_{i}^{new}\left|\right|T_{CS})$, $A_{i}^{new}=h(PID_{i}^{new}\left||MSK\right)$, $M_{7}=A_{i}^{new}⊕h(A_{i}||PID_{i}^{new})$, and $M_{8}=h(A_{i}^{new}\left|\right|r_{1})$, where $\alpha_{j}$ is retrieved from list $L_{V_{j}}$ and $T_{CS}$ is the current timestamp of CS. Finally CS sends the message {$M_{4},M_{5},M_{6},M_{7},M_{8},T_{CS}$} to $V_{j}$ through a public channel.**Step 4.** After receiving the message from CS, $V_{j}$ checks whether $time-T_{CS}\leq \Delta_{T}$ holds or not. If not, $V_{j}$ rejects this session. Otherwise, $V_{j}$ retrieves $\alpha_{j}$ and computes $r_{1}^{\prime }=M_{4}⊕h(\alpha_{j}\left|\right|T_{CS})$, $PID_{i}^{new}=M_{5}⊕h(\alpha_{j}\left|\right|r_{1}^{\prime }\left|\right|T_{CS})$, and $M_{6}^{\prime }=h(\alpha_{j}\left|\right|r_{1}^{\prime }||PID_{i}^{new}\left|\right|T_{CS})$.**Step 5.** $V_{j}$ checks whether $M_{6}^{\prime }=M_{6}$ holds or not. If not, $V_{j}$ rejects the request. Otherwise, $V_{j}$ authenticates the legality of CS and $U_{i}$. Then, $V_{j}$ randomly chooses a 160 bits random number $r_{2}\in Z_{n}^{*}$ and computes the common session key $SK_{ji}=h\left(PID_{i}^{new}⊕r_{1}^{\prime }⊕r_{2}\right)$, $M_{9}=h(PID_{i}^{new}\left|\right|r_{1}^{\prime })⊕r_{2}$, and $M_{10}=h(PID_{i}^{new}\left|\right|r_{1}^{\prime }\left|\right|r_{2}||SK_{ji}\left|\right|T_{V_{j}})$, where $T_{V_{j}}$ is the current timestamp of $V_{j}$. Finally $V_{j}$ sends the message {$M_{7},M_{8},M_{9},M_{10},T_{V_{j}}$} to $U_{i}$ through a public channel.**Step 6.** After receiving the message from $V_{j}$, $U_{i}$ checks whether $time-T_{V_{j}}\leq \Delta_{T}$ holds or not. If not, $U_{i}$ rejects this session. Otherwise, $U_{i}$ computes $A_{i}^{\prime new}=M_{7}⊕h(A_{i}||PID_{i}^{new})$ and $M_{8}^{\prime }=h(A_{i}^{\prime new}\left|\right|r_{1})$. Then $U_{i}$ further checks if $M_{8}^{\prime }=M_{8}$ holds or not. If it is true, it implies that CS is authenticated to $U_{i}$. In order to verify the legality of $V_{j}$, $U_{i}$ computes $r_{2}^{\prime }=M_{9}⊕h(PID_{i}^{new}\left|\right|r_{1})$, the common session key $SK_{ij}=h\left(PID_{i}^{new}⊕r_{1}⊕r_{2}^{\prime }\right)$, and $M_{10}^{\prime }=h(PID_{i}^{new}\left|\right|r_{1}\left|\right|r_{2}^{\prime }||SK_{ij}\left|\right|T_{V_{j}})$ and checks whether $M_{10}^{\prime }=M_{10}$ holds or not. If not, $U_{i}$ rejects the communication request. Otherwise, it implies that $V_{j}$ is also authenticated to $U_{i}$ and the common session key $SK_{ij}=h\left(PID_{i}^{new}⊕r_{1}⊕r_{2}\right)=SK_{ji}$ will be used for securing IoD communications between $U_{i}$ and $V_{j}$. Finally, $U_{i}$ computes $B_{i}^{new}=A_{i}^{new}⊕h\left(PW_{i}\right)$ and replaces {$B_{i},r_{U_{i}}$} with {$B_{i}^{new},r_{U_{i}}^{new}$} for the next login.

## 4. Security Analysis of the Proposed Scheme

In this section, meticulous informal security analysis and the security verification are carried out using ProVerif to prove the security and the validity of the proposed scheme. In addition, BAN logic is utilized to corroborate the logical exactitude of the proposed scheme.

### 4.1. Simulation Verification with ProVerif

ProVerif is a proper tool that can automatically analyze cryptographic protocols and verify the security and reliability of authentication protocols. The specific operation of ProVerif is described in detail below.

The symbols used in the proof process are defined as shown in Figure 5. The “sch” and “ch” refer to the secure channel and the common channel. The functions used mainly include h(),xor(), and con(), which represent the hash operation, XOR operation and join operation, respectively. Figure 6 shows the defined queries and events. Here, $SK_{ij}$ and $SK_{ji}$ represent the common session keys of the user and the drone, respectively. The event UserStarted() indicates that the user $U_{i}$ starts working, the event UserAuthed() indicates that the user is authenticated, the event ControlServerAcUser() indicates that the control server CS authenticates the user event, the event DroneAcControlServer() indicates that the drone $V_{j}$ authenticates the control server event, the event UserAcControlServer() indicates that the user $U_{i}$ authenticates the control server event, and the event UserAcDrone() means that the user $U_{i}$ authenticates the drone event.

The tripartite agreement of user $U_{i}$, drone $V_{j}$ and control server CS are converted into ProVerif code as shown in Figure 7, Figure 8 and Figure 9, respectively. In the working process of $U_{i}$, out(sch,(IDi,PIDi)) and in(sch,(xAi:bitstring)) represent the messages sent and received by $U_{i}$ through the secure channel during the registration phase. After completing the registration, $U_{i}$ starts authentication by executing the event UserStarted(). Next, out(ch,(PIDi,M1,M2,M3,TUi)) represents the message is transmitted from $U_{i}$ to CS over the common channel, in(ch,(xM7:bitstring,xM8:bitstring,xM9:bitstring,xM10:bitstring,xTVj:bitstring)) represents the message is transmitted from $V_{j}$ to $U_{i}$ over the common channel. In addition, the working process of CS includes UiReg for $U_{i}$ registration by CS, VjReg for $V_{j}$ registration by CS, and CSAuth means the authentication operation of CS.

Finally, the results of the execution of the ProVerif code are shown in Figure 10. Based on the results of Figure 10, it shows that the sequence of events is normal and it can be proved that the attacker cannot derive the common session key shared among $U_{i}$ and $V_{j}$ during IoD communications.

### 4.2. BAN Logic Analysis

In the proposed scheme, when the mobile device wants to communicate with the flying drone, they must authenticate each other. In the following description, we use the BAN logic model to prove the security of the proposed scheme. The notation of BAN logic is described as follows:**-**P|≡X:*P* believes *X* or *P* would be entitled to believe *X*.**-**P⊲X:*P* sees *X*. Someone has sent a message containing *X* to *P*, who can read and repeat *X*.**-**P|⇒X:*P* has jurisdiction over *X*. *P* is an authority on *X* and should be trusted on this matter.**-**P|∼X:*P* once said *X*. *P* at some time sent a message including *X*.**-**$<X{>}_{Y}$:This represents *X* combined with *Y*.**-**♯(X):The formula *X* is fresh, that is, *X* has not been sent in a message at any time before the current run of the protocol.**-**$P\begin{array}{c} K\\ ⟷ \end{array}Q$:*P* and *Q* may use the shared key *K* to communicate.**-**$P\begin{array}{c} S\\ ⟺ \end{array}Q$:The formula *S* is a secret known only to *P* and *Q* and possibly to principals trusted by them.

In the authentication and key-agreement phase of the proposed scheme, the main goal of our scheme is to authenticate the session key establishment between a mobile user $U_{i}$ and the flying drone $V_{j}$.

**G1:** 

$U_{i}\;|\equiv U_{i}\begin{array}{c} SK_{ij}\\ ⟷ \end{array}V_{j}$

**G2:** 

$U_{i}\;\left|\equiv V_{j}\;\right|\equiv U_{i}\begin{array}{c} SK_{ij}\\ ⟷ \end{array}V_{j}$

**G3:** 

$V_{j}\;|\equiv U_{i}\begin{array}{c} SK_{ij}\\ ⟷ \end{array}V_{j}$

**G4:** 

$V_{j}\;\left|\equiv U_{i}\;\right|\equiv U_{i}\begin{array}{c} SK_{ij}\\ ⟷ \end{array}V_{j}$

**G5:** 

$V_{j}\;|\equiv ID_{i}$

**G6:** 

$V_{j}\;\left|\equiv U_{i}\;\right|\equiv ID_{i}$



According to the authentication and key agreement phase, we use BAN logic to produce an idealized form as follows:**M1:** $\left(<PID_{i}{>}_{h(ID_{i}||PW_{i}||r_{U_{i}})},<r_{1}{>}_{h\left(PID_{i}\right)⊕r_{1}},<U_{i}\begin{array}{c} SK_{ij}\\ ⟷ \end{array}V_{j}{>}_{h(PID_{i}⊕r_{1}⊕r_{2})}\right)$**M2:** $\left(<r_{2}{>}_{h(PID_{i}\left|\right|r_{1})⊕r_{2}},<U_{i}\begin{array}{c} SK_{ij}\\ ⟷ \end{array}V_{j}{>}_{h(PID_{i}⊕r_{1}⊕r_{2})}\right)$

To analyze the proposed scheme, we make the following assumptions:**A1:** $U_{i}\;|\equiv ♯\left(PID_{i}\right)$**A2:** $V_{j}\;|\equiv ♯\left(PID_{i}\right)$**A3:** $U_{i}\;|\equiv U_{i}\begin{array}{c} h(PID_{i}⊕r_{1}⊕r_{2})\\ ⟺ \end{array}V_{j}$**A4:** $V_{j}\;|\equiv U_{i}\begin{array}{c} h(PID_{i}⊕r_{1}⊕r_{2})\\ ⟺ \end{array}V_{j}$**A5:** $U_{i}\;\left|\equiv V_{j}\;\right|\Rightarrow U_{i}\begin{array}{c} SK_{ij}\\ ⟷ \end{array}V_{j}$**A6:** $V_{j}\;\left|\equiv U_{i}\;\right|\Rightarrow U_{i}\begin{array}{c} SK_{ij}\\ ⟷ \end{array}V_{j}$**A7:** $V_{j}\;\left|\equiv U_{i}\;\right|\Rightarrow ID_{i}$

According to these assumptions and rules of BAN logic, we show the main proof of the session key establishment between a mobile user $U_{i}$ and the flying drone $V_{j}$ as follows:

Flying drone $V_{j}$ authenticates mobile device $U_{i}$. By **M1** and the *seeing rule*, we can derive:**S1:** $V_{j}\;⊲\;\left(<PID_{i}{>}_{h(ID_{i}||PW_{i}||r_{U_{i}})},<r_{1}{>}_{h\left(PID_{i}\right)⊕r_{1}},<U_{i}\begin{array}{c} SK_{ij}\\ ⟷ \end{array}V_{j}{>}_{h(PID_{i}⊕r_{1}⊕r_{2})}\right)$

By **A2** and the *freshness rule*, we can derive:**S2:** $V_{j}\;|\equiv ♯\left(<PID_{i}{>}_{h(ID_{i}||PW_{i}||r_{U_{i}})},<r_{1}{>}_{h\left(PID_{i}\right)⊕r_{1}},<U_{i}\begin{array}{c} SK_{ij}\\ ⟷ \end{array}V_{j}{>}_{h(PID_{i}⊕r_{1}⊕r_{2})}\right)$

By **S1**, **A4** and the *message meaning rule*, we can derive:**S3:** $V_{j}\;\left|\equiv U_{i}\;\right|∼\left(<PID_{i}{>}_{h(ID_{i}||PW_{i}||r_{U_{i}})},<r_{1}{>}_{h\left(PID_{i}\right)⊕r_{1}},<U_{i}\begin{array}{c} SK_{ij}\\ ⟷ \end{array}V_{j}{>}_{h(PID_{i}⊕r_{1}⊕r_{2})}\right)$

By **S2**, **S3**, and the *nonce verification rule*, we can derive:**S4:** $V_{j}\;\left|\equiv U_{i}\;\right|\equiv \left(<PID_{i}{>}_{h(ID_{i}||PW_{i}||r_{U_{i}})},<r_{1}{>}_{h\left(PID_{i}\right)⊕r_{1}},<U_{i}\begin{array}{c} SK_{ij}\\ ⟷ \end{array}V_{j}{>}_{h(PID_{i}⊕r_{1}⊕r_{2})}\right)$

By **S4** and the *belief rule*, we can derive:**S5:** $V_{j}\;\left|\equiv U_{i}\;\right|\equiv U_{i}\begin{array}{c} SK_{ij}\\ ⟷ \end{array}V_{j}$

By **S5**, **A6** and the *jurisdiction rule*, we can derive:**S6:** $V_{j}\;|\equiv U_{i}\begin{array}{c} SK_{ij}\\ ⟷ \end{array}V_{j}$

By **S6** and the *belief rule*, we can derive:**S7:** $V_{j}\;\left|\equiv U_{i}\;\right|\equiv ID_{i}$

By **S7**, **A7** and the *jurisdiction rule*, we can derive:**S8:** $V_{j}\;|\equiv ID_{i}$

Mobile device $U_{i}$ authenticates flying drone $V_{j}$. By **M2** and the *seeing rule*, we can derive:**S9:** $U_{i}◃\left(<r_{2}{>}_{h(PID_{i}\left|\right|r_{1})⊕r_{2}},<U_{i}\begin{array}{c} SK_{ij}\\ ⟷ \end{array}V_{j}{>}_{h(PID_{i}⊕r_{1}⊕r_{2})}\right)$

By **A1** and the *freshness rule*, we can derive:**S10:** $U_{i}\;|\equiv ♯\left(<r_{2}{>}_{h(PID_{i}\left|\right|r_{1})⊕r_{2}},<U_{i}\begin{array}{c} SK_{ij}\\ ⟷ \end{array}V_{j}{>}_{h(PID_{i}⊕r_{1}⊕r_{2})}\right)$

By **S9**, **A3** and the *message meaning rule*, we can derive:**S11:** $U_{i}\;\left|\equiv U_{i}\;\right|∼\left(<r_{2}{>}_{h(PID_{i}\left|\right|r_{1})⊕r_{2}},<U_{i}\begin{array}{c} SK_{ij}\\ ⟷ \end{array}V_{j}{>}_{h(PID_{i}⊕r_{1}⊕r_{2})}\right)$

By **S10**, **S11**, and the *message meaning rule*, we can derive:**S12:** $U_{i}\;\left|\equiv V_{j}\;\right|\equiv U_{i}\begin{array}{c} SK_{ij}\\ ⟷ \end{array}V_{j}$

By **S12**, **A5**, and the *jurisdiction rule*, we can derive:**S13:** $U_{i}\;|\equiv U_{i}\begin{array}{c} SK_{ij}\\ ⟷ \end{array}V_{j}$

By **S5**, **S8**, **S12** and **S13**, it can be proved that, in our authentication scheme, the mobile device $U_{i}$ and the flying drone $V_{j}$ authenticate each other with the help of control server *CS*. In addition, we are also able to prove that the proposed scheme can establish a common session key *SK_ij_* between the mobile device *U_i_* and the remote flying drone *V_j_* with the help of *CS*. Finally, the authentication and key agreement phase of our scheme thus guarantee the security of $SK_{ij}$ between $U_{i}$ and $V_{j}$.

**Scenario:** A malicious attacker uses an illegal flying drone $V_{j}$ to authenticate a legal mobile device $U_{i}$.**Analysis:** The attacker will not succeed because the illegal flying drone $V_{j}$ has not been registered to the legal control server CS, and the illegal flying drone $V_{j}$ cannot calculate the correct session key SK. Thus, it will fail when the legal mobile device $U_{i}$ attempts to authenticate the illegal flying drone $V_{j}$. In the proposed scheme, the attacker cannot achieve their purpose using an illegal flying drone $V_{j}$. In the same scenario, the proposed scheme can also defend against a malicious attack using an illegal mobile device $U_{i}$ to connect to a legal flying drone $V_{j}$. This is because the illegal mobile device $U_{i}$ has not been registered to the legal control server CS, and thus the illegal mobile device $U_{i}$ cannot calculate the correct session key *SK*. Therefore, the attack will fail when the legal flying drone $V_{j}$ attempts to authenticate the illegal mobile device $U_{i}$.

### 4.3. Informal Security Analysis

In this subsection, we present the informal security analysis of the proposed scheme and show it can satisfy the following security features and attack resilience in IoD environments.

**Proposition** **1.**
*The proposed scheme ensures anonymous interactions between $U_{i}$, CS and $V_{j}$ and no adversaries can ascribe any session to a particular user during authentication and key agreement phase.*


**Proof.** According to DY threat model defined in Section 2.2, an adversary A can collect all the communication messages transmitted in IoD, such as $\left\{PID_{i},M_{1},M_{2},M_{3},T_{U_{i}}\right\}$, {$M_{4},M_{5},M_{6},M_{7},M_{8},T_{CS}$}, and {$M_{7},M_{8},M_{9},$$M_{10},T_{V_{j}}$}, which are communicated during the authentication and key agreement phase of the proposed scheme. From these messages, it is hard for A to derive $U_{i}$’s real identity $ID_{i}$ from $PID_{i}$ without knowing the random number $r_{U_{i}}$ because $PID_{i}$ is protected with cryptographic hash function h(·). That is to say, $U_{i}$’s real identity are transmitted in cipher format instead of plaintext. Therefore, the user anonymity can be provided in the proposed authentication scheme. □

**Proposition** **2.**
*The proposed scheme ensures untraceability between a mobile user and the control server and also between a mobile and its associated drone.*


**Proof.** In the proposed authentication mechanism, the generation of messages {$PID_{i},M_{1},$$M_{2},M_{3},M_{4},M_{5},M_{6},M_{7},M_{8},M_{9},M_{10}$} incorporate the fresh random numbers $r_{U_{i}}$, $r_{1}$, and $r_{2}$ and the pseudonym ID and session key is updated after each successful authentication. As a result, it is impossible for A to correlate the communicated messages from the current and previous AKA process and the proposed scheme can provide untraceability. □

**Proposition** **3.**
*The proposed scheme supports mutual authentication between any two communicating parties, and also between a drone $V_{j}$ and its associated $U_{i}$.*


**Proof.** During the proposed authentication process as presented in Section 3.4, a drone $V_{j}$ verifies its associated $U_{i}$’s legitimacy before establishment of a session key. In the session, CS first checks the freshness of $U_{i}$’s login request by validating the timestamp $T_{U_{i}}$ in the messages $\left\{PID_{i},M_{1},M_{2},M_{3},T_{U_{i}}\right\}$. Later, CS checks $M_{3}$ to authenticate $U_{i}$. When receiving {$M_{4},M_{5},M_{6},M_{7},$$M_{8},T_{CS}$} from CS, $V_{j}$ checks $T_{CS}$ and $M_{6}$ to authenticate CS and $U_{i}$. If both the conditions are validated successfully, $V_{j}$ agrees a session key with $U_{i}$. In the similar way, when receiving {$M_{7},M_{8},M_{9},M_{10},T_{V_{j}}$}, $U_{i}$ checks $T_{V_{j}}$ and $M_{10}$ to authenticate $V_{j}$ and $U_{i}$ also agrees a session key with $V_{j}$. Finally, the proposed scheme achieves mutual authentication and both $U_{i}$ and $V_{j}$ ensure that they shared the same session key with the help of CS for securing the future IoD communications. □

**Proposition** **4.**
*The proposed scheme is secure against session key exposure attack.*


**Proof.** After the successful authentication process, $U_{i}$ and $V_{j}$ can establish a common session key $SK_{ij}=h\left(PID_{i}^{new}⊕r_{1}⊕r_{2}\right)$ and the adversary A may try to derive $SK_{ij}$ to damage the later IoD communications between them. However, in **Step 1** of the authentication and key agreement phase, A cannot get $PID_{i}^{new}$ and $r_{1}$ from $M_{1}$ and $M_{2}$ without knowing the knowledge of $A_{i}=h(PID_{i}\left||MSK\right)$. Similarity, in **Step 5** of the authentication and key agreement phase, A cannot obtain $r_{2}$ from $M_{9}$ without knowing the knowledge of $\alpha_{j}=h(ID_{j}\left||k\right)$. Therefore, A cannot get success from session key disclosure attack in the proposed AKA scheme. □

**Proposition** **5.**
*The proposed scheme is resilient against known session key attack.*


**Proof.** It can be observed from Section 3.4 that the session key $SK_{ij}$ is the combination of both session-specific credential $PID_{i}\left[new\right]$ and two 160 bits random numbers $r_{1}$ and $r_{2}$. Moreover, usage of session-specific credentials and random numbers in computation of session keys between $U_{i}$ and $V_{j}$ over different sessions make always-unique session keys. Even if a session key is disclosed for a specific session, it will not result in computing the session keys over other sessions. Thus, the contributed scheme is protected from known session key attack. □

**Proposition** **6.**
*The proposed scheme is protected against drone capture attack.*


**Proof.** According to CK adversary model defined in Section 2.2, an adversary A may physically capture the drone in the sensing environment and maliciously extract the stored contents from its memory by using power analysis attacks. In this way, A can get {$ID_{j},\alpha_{j}$} from the memory of compromised drone $V_{j}^{\prime }$. By capturing $V_{j}^{\prime }$, A can only compromise the session key between a victim user $U_{i}$ and $V_{j}^{\prime }$. Since all the identities and credentials for all $V_{j}$ are distinct in IoD network, A cannot compromise other non-captured drone due to the distinct as well as uniqueness property of the contents stored in the remote drones. Finally, compromise of a drone does not result in damaging secure IoD communications among a user and other non-compromised drones and the contributed scheme is resilient against drone capture attack. □

**Proposition** **7.**
*The proposed scheme is secure against stolen device attack.*


**Proof.** Suppose an adversary A somehow gets or steals the mobile device of user $U_{i}$ and extracts the stored contents {$B_{i},r_{U_{i}}$} from its memory by using power analysis attacks. Thus, A can get access to IoD environment. However, A cannot drive the valid secret credential $A_{i}$ due to the protection of $U_{i}$’s password. Moreover, the password is protected in the form of a one-way hash function which is a non-invertible function. Although A can guess the password of $U_{i}$, he/she cannot verify the correctness without having $U_{i}$’s identity $ID_{i}$ and the login parameters of previous session. Therefore, the contributed scheme can resist stolen device attack. □

**Proposition** **8.**
*The proposed scheme is resilient secure against three kinds of impersonation attacks, including: user impersonation, CS impersonation and drone impersonation.*


**Proof.** The following impersonation attacks related to the contributed scheme are taken into account.
**(a)** **User impersonation attack:** Let an adversary A try to behave himeself/herself as a legitimate user $U_{i}$ and he/she wants to generate an authorized login request, say $\left\{PID_{i},M_{1},M_{2},M_{3},T_{U_{A}}\right\}$. A can intercept the login request $\left\{PID_{i},M_{1},M_{2},M_{3},T_{U_{i}}\right\}$ of $U_{i}$ and forge messages by extracting the important credential $PID_{i}$ of $U_{i}$ to prove A’s authenticity. In order to perform this operation, A needs to choose two random numbers $r_{A}^{new}$ and $r_{1}^{*}$ and a timestamp $T_{U_{A}}$ and computes $M_{1}=h\left(r_{A}^{new}\right)⊕h(A_{i}\left|\right|T_{U_{A}})$, $M_{2}=h\left(h\left(r_{A}^{new}\right)\right)⊕r_{1}^{*}$ and $M_{3}=h(h\left(r_{A}^{new}\right)\left|\right|r_{1}^{*}\left|\right|T_{U_{A}})$. However, due to the lack of knowledge about $A_{i}$, A will fail to compute $M_{1}$ as valid login parameter. Therefore, the proposed scheme is secure against user impersonation attack.**(b)** CS**impersonation attack:** To perform this attack, we assume A intercepts the message {$M_{4},M_{5},M_{6},M_{7},M_{8},T_{CS}$} and generates a bogus message {$M_{4}^{*},M_{5}^{*},M_{6}^{*},M_{7}^{*},M_{8}^{*},T_{U_{A}}$} to the drone $V_{j}$, to make $V_{j}$ and $U_{i}$ convince the message is from a legitimate CS, where $T_{U_{A}}$ is a timestamp generated by A. However, A does not have the knowledge of $\alpha_{j}$ and $PID_{i}^{new}$, thus, $V_{j}$ and $U_{i}$ can distinguish the impersonated CS from real control server and the proposed scheme is secure against CS impersonation attack.**(b)** **Drone impersonation attack:** In this attack, A will try to make believe $U_{i}$ by seizing the message {$M_{4},M_{5},M_{6},M_{7},M_{8},T_{CS}$} and attempt to construct another legitimate message, which is authenticated to $U_{i}$. First, A randomly chooses a random number $r_{2}^{*}$ and a timestamp $T_{U_{A}}$ and tries to forge $M_{9}$ and $M_{10}$. However, in the design process of the proposed AKA scheme, without having the knowledge of $\alpha_{j}$, $r_{1}$ and $PID_{i}^{new}$, A cannot generate the valid convinced response to impersonate as an accurate drone.
□

## 5. Performance Evaluation

This section shows a detailed comparison among the proposed scheme and those of the most relevant state-of-the-art schemes in the IoD environment, such as the schemes of Singh et al. [30] and Zhang et al. [29] in terms of security features, computational and communication overheads.

### 5.1. Comparison of Security Features

We highlight on the comparison of security features and attacks protection of the contributed scheme against relevant schemes [29,30] in this section. It is clear from the Table 1 that the scheme of Singh et al. [30] is insecure against session key exposure attack, impersonation attack, drone capture attack and stolen device attack and Zhang et al. [29] is unprotected against session key exposure attack and impersonation attack. Furthermore, the scheme of Singh et al. [30] lacks mutual authentication, user anonymity and untraceability and Zhang et al. [29] does not provide user anonymity and untraceability. Therefore, the proposed AKA scheme can provide more security features and protect against all kinds of attack which makes it more suitable for generic secure communications in IoD-based environments.

### 5.2. Comparison of Computational Overhead

In order to provide the analysis of the comparative computation overhead, the symbols listed in Table 2 with their executing time as per the experiment presented in [31] on a mobile (drone) device with 2.45 G processor and 2 GB memory, performed on the Android 4.4.2 operation system. The control server is simulated on a PC I5-4460S with 2.90 GHz processor and 4 GB memory, performed on Window 8 operation system:

As shown in Table 3, total computational overhead of the proposed scheme, the scheme of Singh et al. [30], Zhang et al. [29] is 27$T_{h}\approx$ 1.022 ms, 4$T_{exp}$+12$T_{mul}\approx$ 9.092 ms, 24$T_{h}\approx$ 1.001 ms, respectively. The computational overhead of the proposed scheme is slightly higher than Zhang et al., whereas the proposed scheme has less computational overhead as compared with the scheme of Singh et al. Moreover, the proposed scheme is more secure than the all rest of the related schemes as proved earlier.

### 5.3. Comparison of Communication Overhead

This section presents another significant performance factor, namely communication overhead, to demonstrate the efficiency of the proposed scheme. For comparison purposes and to keep simplicity, let |G| denote the 1024 bits length of element in *G* and $|Z_{n}|$ denote the 160 bits length of the element in $Z_{n}$. The symbol |T| denotes a timestamp 32 bits in lengts and participant identities. We compare the communication overhead of different participants during the login and authentication phases, where the bits sent over communication channel and the number of messages transmitted between them are also considered.

As shown in Table 4, the number of transmitted messages in Singh et al. scheme are $X_{i},Y_{i},Time_{i},ID_{i}$ from user side and $X_{j},Y_{j},Time_{j},ID_{j}$ from drone side, where $\left(Time_{i},Time_{j}\right)$ are 32-bit timestamps and $\left(ID_{i},ID_{j}\right)$ are 32-bit user identities. Therefore, the total communication cost of Singh et al. scheme is 4|G|+4|T| about 4256 bits. In addition, the number of transmitted messages in Zhang et al. scheme are $M_{1},M_{2},M_{3},M_{4},T_{1}$ from user side, $M_{5},M_{6},M_{7}$ from control server side and $M_{8},M_{10}$ from drone side, where $M_{i}\in Z_{n}$ and $T_{1}$ is a 32-bit timestamp. Thus the total communication cost of Zhang et al. scheme is 9$|Z_{n}|$+|T| about 1472 bits. In the proposed scheme, three message transmissions complete the authentication and key agreement process: (1) user sends {$PID_{i},M_{1},M_{2},M_{3},$$T_{U_{i}}$} to CS; this consumes {160 + 160 + 160 + 160 + 32} = 672 bits, and (2) control server sends {$M_{4},M_{5},M_{6},M_{7},M_{8},T_{CS}$} to $V_{j}$, consuming {160 + 160 + 160 + 160 + 160 + 32} = 832 bits, and (3) drone sends {$M_{7},M_{8},M_{9},M_{10},T_{V_{j}}$} to $U_{i}$, which also needs {160 + 160 + 160 + 160 + 32} = 672 bits. Therefore, the total communication cost of the proposed scheme is 13$|Z_{n}|$+3|T|, about 2176 bits.

## 6. Conclusions

In this paper, we proposed a lightweight hash-based authenticated key agreement and privacy preservation scheme without using symmetric/asymmetric cryptographic operations for IoD environments. The proposed scheme is a three-party AKA mechanism, which enables mobile users to communicate securely, through the public communication channel, with the IoD participants such as control server and drones. Moreover, the proposed scheme can provide anonymity and untraceability of the participants in IoD. We proved the security of the proposed scheme formally through the ProVerif tool and BAN logic analysis as well as informally. The comparative analysis depicts that the proposed scheme achieves better trade-off among security features, computational overhead and communication cost. From the results, it is concluded that the proposed scheme not only supports more security features but is also suitable for the drones or resource-constrained sensing devices in the IoD environments.

## Figures and Tables

**Figure 1 sensors-22-09534-f001:**
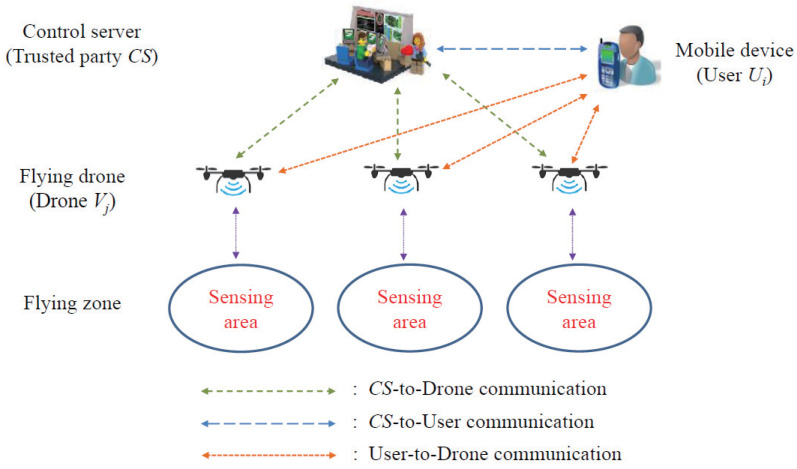
Communication architecture diagram of IoD.

**Figure 2 sensors-22-09534-f002:**
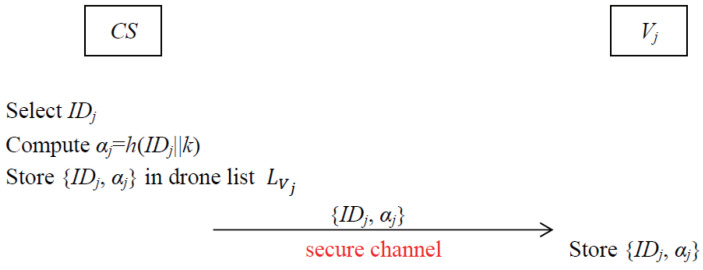
Registration procedure of user.

**Figure 3 sensors-22-09534-f003:**
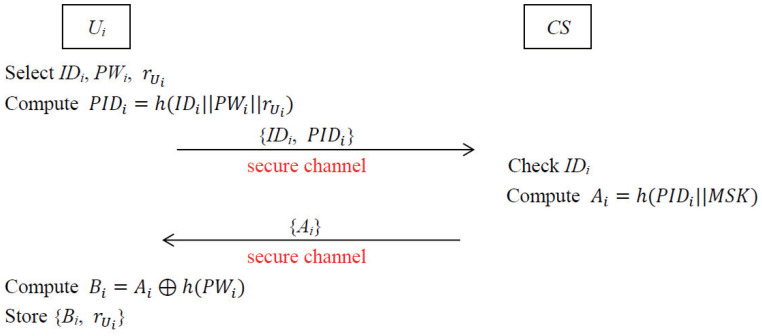
Registration procedure of drone.

**Figure 4 sensors-22-09534-f004:**
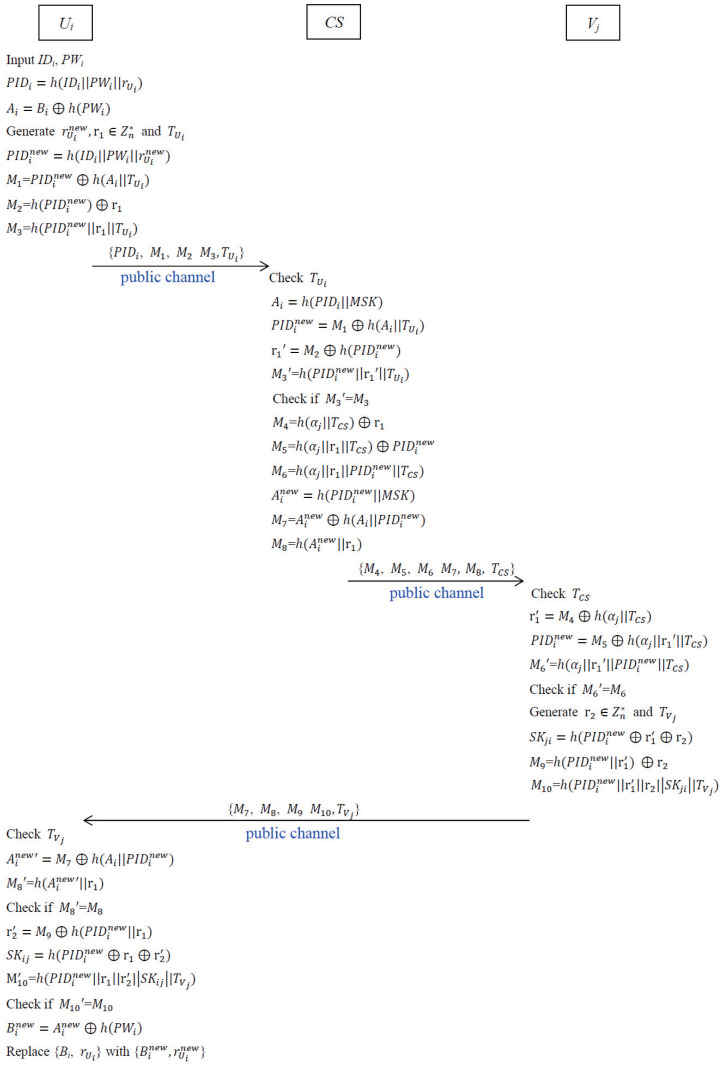
Authentication and key agreement procedure of IoD communications.

**Figure 5 sensors-22-09534-f005:**
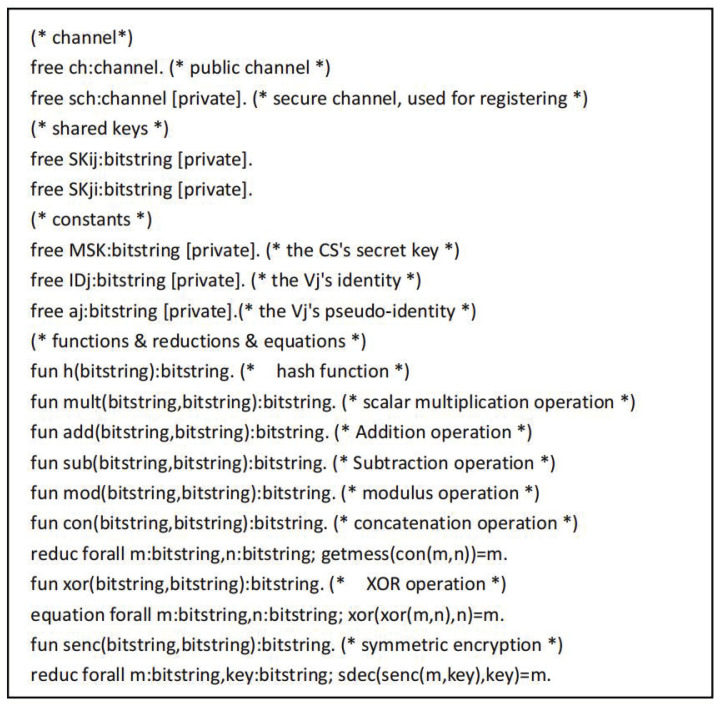
The definition of the proposed protocol in the ProVerif tool.

**Figure 6 sensors-22-09534-f006:**
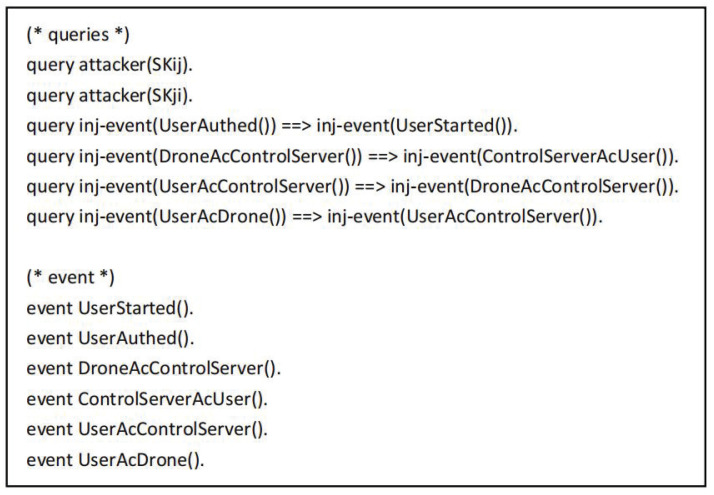
The queries and events in the ProVerif tool.

**Figure 7 sensors-22-09534-f007:**
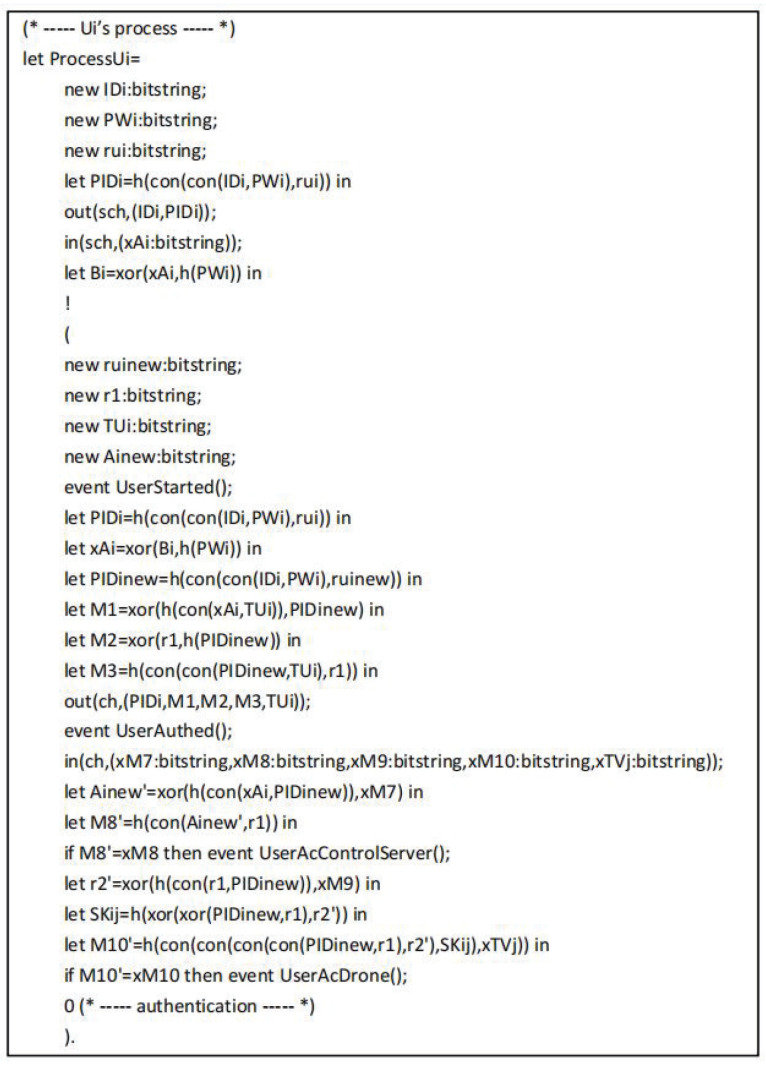
The process of $U_{i}$.

**Figure 8 sensors-22-09534-f008:**
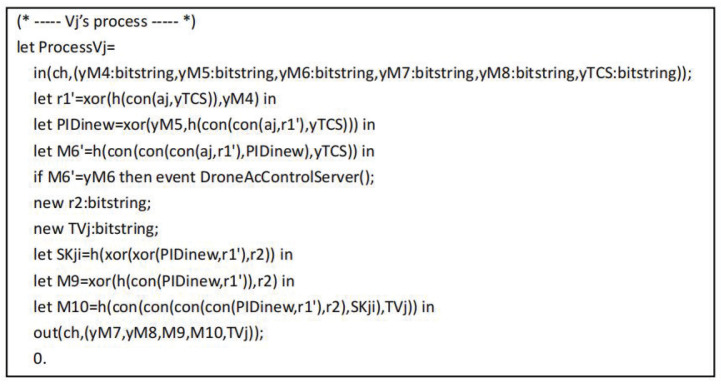
The process of $V_{j}$.

**Figure 9 sensors-22-09534-f009:**
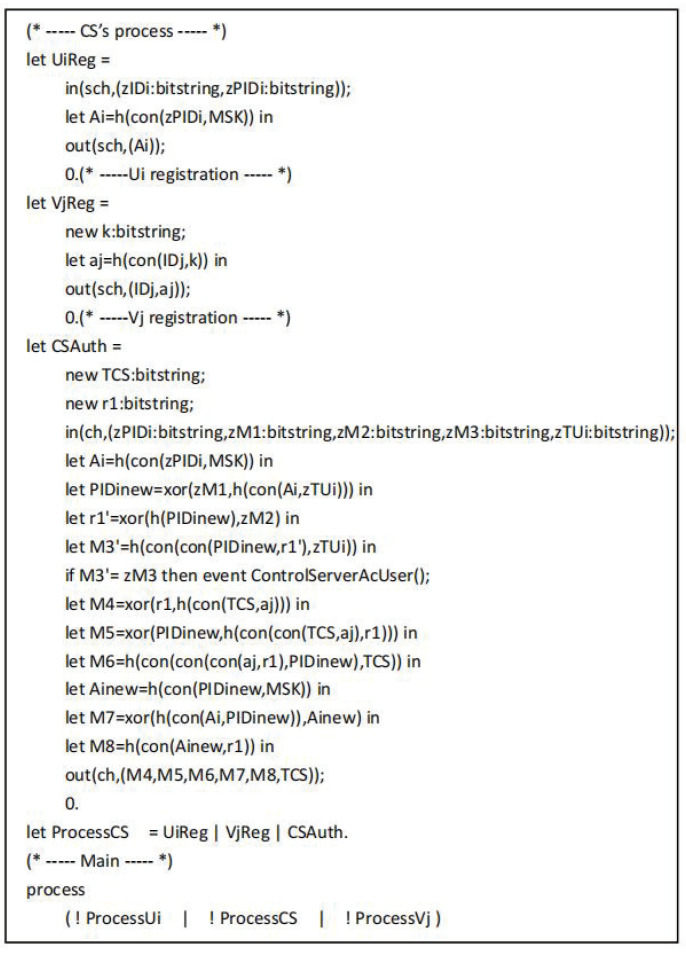
The process of CS.

**Figure 10 sensors-22-09534-f010:**
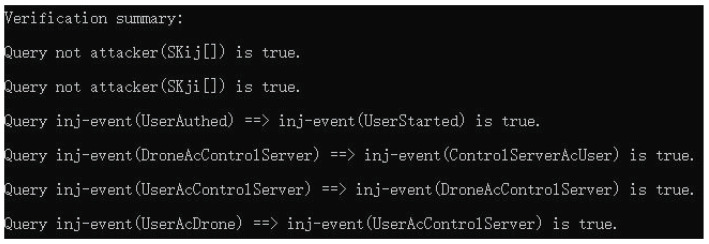
ProVerif results.

**Table 1 sensors-22-09534-t001:** Comparison of security features.

Security Features	Singh et al. [30]	Zhang et al. [29]	Proposed
	(2019)	(2020)	Scheme
Provision of mutual authentication	No	Yes	Yes
Provision of user anonymity	No	No	Yes
Provision of untraceability	No	No	Yes
Prevention of session key exposure attack	No	No	Yes
Prevention of known session key attack	Yes	Yes	Yes
Prevention of replay attack	Yes	Yes	Yes
Prevention of impersonation attack	No	No	Yes
Prevention of drone capture attack	No	Yes	Yes
Prevention of stolen device attack	No	Yes	Yes

**Table 2 sensors-22-09534-t002:** Execution time of the various cryptographic operations.

Symbol	Description	User (Drone) Side	Server Side
$T_{exp}$	Modular exponentiation	2.249 ms	0.339 ms
$T_{mul}$	Modular multiplication	0.008 ms	0.001 ms
$T_{h}$	Secure hash function	0.056 ms	0.007 ms

**Table 3 sensors-22-09534-t003:** Comparison of Computational Overhead.

	Singh et al. [30]	Zhang et al. [29]	Proposed Scheme
	(2019)	(2020)	
User side	2$T_{exp}$ + 5$T_{mul}$	10$T_{h}$	11$T_{h}$
Drone side	2$T_{exp}$ + 7$T_{mul}$	7$T_{h}$	10$T_{h}$
Server side	-	7$T_{h}$	6$T_{h}$
Total	9.092 ms	1.001 ms	1.022 ms

**Table 4 sensors-22-09534-t004:** Comparison of Communication Overhead.

	Singh et al. [30]	Zhang et al. [29]	Proposed Scheme
	(2019)	(2020)	
No. of messages	2	3	3
Communication cost	4|G|+4|T|	9$|Z_{n}|$+|T|	13$|Z_{n}|$+3|T|
Bits length	4256	1472	2176

## Data Availability

Not applicable.

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
