# Peer review of "An Efficient Authenticated Key Agreement Scheme Supporting Privacy-Preservation for Internet of Drones Communications"

_sensors, 2022, doi:10.3390/s22239534_

Round 1
Reviewer 1 Report
In this article, the authors attempted to design a lightweight hash-based authentication that is a very important cryptographic tool to ensure efficiency and confidentiality in the current IoDs. The proposed authentication mechanism is a provably secure with Proverif and BAN logic. They make the authentication much efficient as compared to other asymmetric cryptosystems. The proposed scheme is anonymous and it produces the pseudo-random ciphers. As a result, it helps decrease the false positive rate throughout the authentications and block various security attacks of IoD environments. The authors also embedded nonce-based protocol considering device security, authentication mechanism with mutual authentication, session key generation, the confidentiality of data and data freshness. In general, it is well-written and well-analyzed work made by the authors. A strong point of this work is its appropriate defined formal security model and a sound security analysis. Moreover, the work enhances the security of existing IoD communication systems. The paper is also well-organized. I think the paper is ready for publication. Prior to that, the following minor corrections should be made by the authors:
1. In the section of introduction , the authors should mention the contributions of the proposed scheme.
2. In the discussed threat model, what is the need of CK-adversary model over the DY-model. Please clarify this point.
3. There are some typos in the paper. For example, “preforms” should be replaced with “performs”. All typos and grammar mistakes should be corrected carefully.
Author Response
REVIEWER #1
Comments and Suggestions for Authors
- In the section of introduction , the authors should mention the contributions of the proposed scheme.
Response: We would like to thank Reviewer #1 for his/her valuable comments on the manuscript. To address the concerns raised. We have added one paragraph to mention the contributions of the proposed scheme. Please refer to Page 4, the penultimate paragraph of Section 1.
- In the discussed threat model, what is the need of CK-adversary model over the DY-model. Please clarify this point.
Response: We would like to thank Reviewer #1 for his/her valuable comments on the manuscript. To address the concerns raised. We have mentioned clearly the need of CK-adversary model over the DY-model. Please refer to Page 5, the Section 2.2.
- There are some typos in the paper. For example, “preforms” should be replaced with “performs”. All typos and grammar mistakes should be corrected carefully.
Response: We would like to thank Reviewer #1 for his/her valuable comments on the manuscript. To address the concerns raised. We have invited an English translator to check our paper and we have corrected the related grammar erros or typos in our paper.
Acknowledgement: The authors would like to thank the anonymous referee for their valuable discussions and comments.
Reviewer 2 Report
An efficient authentication and secure communication scheme with privacy preservation is proposed in this paper. The topic is interesting, some following comments can help to revise this paper.
1. There are too many introductions on the application scopes of several drones.
2. There are too many self-cited papers.
3. Theoretical analysis needs to be added.
4. The validation analysis needs to be more detailed.
Author Response
REVIEWER #2
Comments and Suggestions for Authors
- There are too many introductions on the application scopes of several drones.
Response: We would like to thank Reviewer #2 for his/her valuable comments on the manuscript. To address the concerns raised. We have reduced the introductions on the application scopes of several drones.
- There are too many self-cited papers.
Response: We would like to thank Reviewer #2 for his/her valuable comments on the manuscript. To address the concerns raised. We have reduced the self-citation rate of the manuscript.
- Theoretical analysis needs to be added.
Response: We would like to thank Reviewer #2 for his/her valuable comments on the manuscript. To address the concerns raised. We have added more descriptions on theoretical analysis. Please refer to Page 14, the Section 4.2.
4. The validation analysis needs to be more detailed.
Response: We would like to thank Reviewer #2 for his/her valuable comments on the manuscript. To address the concerns raised. We have updated the descriptions on validation analysis. Please refer to Page 9 to Page 12, the Section 4.1.
Acknowledgement: The authors would like to thank the anonymous referee for their valuable discussions and comments.
Round 2
Reviewer 2 Report
This paper has been revised and it can be accepted.